# Pathological and Molecular Diagnosis of Uveal Melanoma

**DOI:** 10.3390/diagnostics14090958

**Published:** 2024-05-02

**Authors:** Mihai Adrian Păsărică, Paul Filip Curcă, Christiana Diana Maria Dragosloveanu, Alexandru Călin Grigorescu, Cosmin Ionuț Nisipașu

**Affiliations:** 1Clinical Department of Ophthalmology, “Carol Davila” University of Medicine and Pharmacy, 020021 Bucharest, Romania; m.pasarica@yahoo.com (M.A.P.); christianacelea@gmail.com (C.D.M.D.); 2Department of Ophthalmology, Clinical Hospital for Ophthalmological Emergencies, 010464 Bucharest, Romania; 3Department of Oncology, Clinical Hospital of Nephrology Dr. Carol Davila, 010731 Bucharest, Romania; alexgrigorescu2004@yahoo.com; 4Department of Dental Medicine I, Implant-Prosthetic Therapy, “Carol Davila” University of Medicine and Pharmacy, 020021 Bucharest, Romania; cosmin.nisipasu@gmail.com

**Keywords:** uveal melanoma, metastatic, oncogene, prognosis, review

## Abstract

(1) Background: Uveal melanoma (UM) is a common malignant intraocular tumor that presents with significant genetic differences to cutaneous melanoma and has a high genetic burden in terms of prognosis. (2) Methods: A systematic literature search of several repositories on uveal melanoma diagnosis, prognosis, molecular analysis, and treatment was conducted. (3) Results: Recent genetic understanding of oncogene-initiation mutations in GNAQ, GNA11, PLCB4, and CYSLTR2 and secondary progression drivers of BAP1 inactivation and SF3B1 and EIF1AX mutations offers an appealing explanation to the high prognostic impact of adding genetic profiling to clinical UM classification. Genetic information could help better explain peculiarities in uveal melanoma, such as the low long-term survival despite effective primary tumor treatment, the overwhelming propensity to metastasize to the liver, and possibly therapeutic behaviors. (4) Conclusions: Understanding of uveal melanoma has improved step-by-step from histopathology to clinical classification to more recent genetic understanding of oncogenic initiation and progression.

## 1. Introduction

Uveal melanoma (UM) is the most common primary malignant intraocular tumor in adults [1] and presents with significant genetic differences to cutaneous melanoma [1]. While the treatment of the primary tumor is well established and successful, given adequate diagnostic timing, the overall survival is darkened by the high long-term risk of metastasis and the poor treatment response of metastatic disease. Rantala E.S. summarized that about half of uveal melanomas (52%) eventually produce clinical metastases by 35 years [1], leading to a 43% mortality rate over 10 years [1]. The most common site for UM metastasis is the liver (over 90%) [1]; synchronous metastases with the primary tumor could be present in 1–3% of cases [1]. The prediction of metastatic risk has evolved from the original Tumor–Node–Metastasis classification (TNM) of the American Joint Committee on Cancer (AJCC) [2,3] to genetic and immunological analysis and, subsequently, The Cancer Genome Atlas (TCGA) classification [3,4]. This classification follows chromosome 3 and 8 mutations, such as disomy 3, monosomy 3, disomy 8q, and 8q gain, and was successfully validated in a large 10-year study reported by Vichitvejpaisal P. et al. at the Wills Eye Hospital in Philadelphia [5], with 5-year cumulative rates for distant metastasis reported at 4% for class A, 20% for class B, 33% for class C, and 63% for class D. Undiagnosed micro-metastases could contribute to almost half of later developed clinical metastases [6]. Current research on UM biology is thus shifting towards addressing metastatic disease. Arising from melanocytes located anywhere in the uveal tract [7], uveal melanoma critically presents both genetic differences and genetic pleomorphism in comparison to cutaneous melanoma, which commonly arises from epidermis basal layer melanocytes [7]. The use of common systemic therapeutic agents for cutaneous melanoma, such as dacarbazine, temozolomide, cisplatin, treosulfan, and fotemustine, has produced lackluster results for the treatment of uveal melanoma [7]. Furthermore, the novel immunomodulatory systemic therapies used successfully for cutaneous melanoma, such as T-lymphocyte-associated antigen 4 (CTLA-4) monoclonal antibody blocker Ipilimumab and Programmed-cell-death-1 (PD-1) inhibitors Nivolumab and Pembrolizumab have also produced uncertain results for uveal melanoma [7,8], with the small-cohort clinical study conducted by Kottschade L.A. et al. reporting varying disease control in half of the patients and progression in the other half [8]. Another avenue, Mitogen-Activated Protein Kinase (MEK) [9] inhibitors, have produced little response regardless of the inhibiting agent and their combination [10]. Overall, despite established improvement in primary tumor treatment and prognosis of metastasis risk, the possibilities of therapeutic intervention remain confounded by the appearance of metastatic disease.

## 2. Materials and Methods

A systematic literature search of several repositories such as the National Library of Medicine’s MEDLINE, Thomson Reuters Scientific Web of Science, and Cochrane Library was performed for uveal melanoma papers published on the subject of diagnosis, prognosis, molecular analysis, and the treatment of uveal melanoma, with a focus on research published in the last 5–10 years. Selected information was grouped accordingly into pathological and molecular diagnoses for uveal melanoma.

## 3. Results

### 3.1. Pathological Diagnosis

Uveal melanoma arises primarily from the choroid (90%) [11] and, less frequently, from other structures anywhere in the uveal tract, such as the iris and ciliary body [11,12]. Approximately 5% of uveal melanomas are isolated to the iris [13].

#### 3.1.1. Primary Tumor Composition

The primary tumor can be constituted of epithelioid or spindle cells [11] and may present predominantly epithelioid cells, spindle cells, or a mixed cell type [11]. Several classifications have been described. Callender GR. et al. described spindle A and B cell types, spindle cells with fascicular patterns similar to schwannomas [14], mixed cell types, pure epithelioid cell types, and too-necrotic-to-classify types [11,14]. Spindle A cells present with an elongated shape, with elongation of the nucleoli, longitudinal nuclear grooves, and few mitoses [11]. Comparatively, spindle B cells are more rounded in shape, with larger nuclei, coarser, clumped chromatin, and small, eccentrically located nucleoli; the cellular borders are more indistinct and, occasionally, mitoses may be observed [11]. Epithelioid cells are more sizeable cells shaped polygonally, with round pleomorphic nuclei, coarse and mostly migrated chromatin, and one or more eosinophilic nucleoli, often paracentral to the nuclei. In comparison to spindle cell types, epithelioid cells have visible cell borders, more cytoplasm, and greater mitotic activity [11]. Larger primary tumors present a higher number of epithelioid cells. Despite the nomenclature of uveal “melanoma”, melanin pigment itself can be minimal or absent in about half of uveal melanomas [11]. When present, melanin pigment is characteristic of epithelioid cells together with the presence of macrophages [11]; tumor necrosis can be present in the epithelioid components of larger tumors.

Tumor organization differs according to the predominance of certain cell types. Spindel A cells present with a syncytial arrangement, Spindel B cells present with both syncytial and fascicular arrangements, and epithelioid cells present with considerable cellular dissociation, with a sheet-like pattern [11].

#### 3.1.2. Primary Tumor Pathological Classifications

The Callender GR classification [11] has been simplified by the Armed Forces Institute of Pathology (AFIP) classification into either exclusively spindle cell tumor types or, if epithelioid cells are present, into the mixed cell type [11,14]. AFIP authors noticed that mixed-cell type tumors comprised of small epithelioid cells had almost as poor a prognosis as mixed-cell type tumors with large epithelioid cells [14] and thus included small epithelioid cells into the mixed-type group [14]. Furthermore, even the sporadic presence of epithelioid cells was associated with worse prognosis [14].

The authors of the AFIP first recommended placing greater importance on nuclear size and pleomorphism in order to better differentiate cells, and second, that with increasing pleomorphism rate, survival time decreased. The presence of epithelioid cells with cytological criteria of malignancy (nuclear pleomorphism and prominent nucleoli) is the differentiating criteria between uveal melanoma and simple uveal tract naevus, with naevi presenting either bland spindle cells or physiological nuclear-aspect epithelioid cells [11].

The American Joint Committee on Cancer (AJCC) classification uses the universal tumor (T), node (N) and metastasis (M) (TNM) staging system for iris melanoma and posterior uveal melanoma involving the ciliary body and choroid [15,16,17]. The 7th edition has been superseded by the 8th edition [18] and will also be superseded by the 9th version, which is currently under construction chapter by chapter [19] and will include several neoplasms, such as cervical cancer [19]. The AJCC has changed from an edition system to a version system and thus the 9th version is to replace previous editions when available for the respective pathology. The 8th classification is currently in use for uveal melanoma [18,20], which is distinct from posterior uveal melanoma (choroidal and ciliary body) and separate from iris melanoma. The 8th AJCC Classification has organized T based on the largest basal diameter and thickness into increasing categories, with subclassifications for ciliary body involvement and extraocular extension [18,20].

#### 3.1.3. Clinical Validation Findings of the American Joint Committee on Cancer (AJCC) Classification and Identification of Further Clinical Data for Prognosis Estimation

Several authors have reported the clinical validation of the AJCC TNM classification via the statistical correlation between larger tumor size (T) or higher staging (S) at diagnosis and increased metastasis rate with decreased survival rates. Table 1 presents a summary of the studies.

Shields CL. et al. performed two studies on a cohort of 7731 patients, first analyzing the significance of diagnosis tumor size (T from TNM) [15] and, afterwards, the impact of TNM staging [21]. The first study in 2013 strongly associated larger tumor sizes (TNM T) with increased metastatic risk and subsequent non-survival (*p* < 0.001) [15] (Table 1). Extra-classification factors were also identified using multivariate analysis: for T1, older age, ciliary body involvement, and increasing tumor base were identified; for T2, older age, multi-quadrant location, dome or plateau configuration, and amelanotic melanoma were identified; for T3, older age, inferior location, close proximity to foveola, increasing tumor base, amelanotic melanoma, and intraocular hemorrhage were identified; and for T4, increasing tumor thickness, plateau configuration, and mixed pigmentation were identified. The second study in 2015 found a similar effect but for TNM staging at diagnosis: compared with Stage I uveal melanoma, Stage II increased the risk of metastasis by 3.1 times and Stage III by 9.3 times [21] (Table 1).

Kujala E. et al.’s analysis validated data for 2152 tumors and reported that survival estimates decreased with increasing stage (Table 1) [22].

A retrospective study conducted by Smidt-Nielsen I. et al. on 3344 Danish patients looked at over-time improvements to AJCC survival rates by extracting data from pathology reports and clinical records over a 70-year period corresponding to the implementation of Danish personal identification number records [23]. These data were subsequently subjected to 8th AJCC uveal melanoma classification [23]. This study is important for reporting improvement in survival compared to the background population in all T categories except for T4 [23], confirming the findings of Bergman L. et al. of improving survival [24]. The Smidt-Nielsen I. et al. study reported decreased metastatic death by 1% for each calendar year of diagnosis for tumor size categories defined by the AJCC [23].
diagnostics-14-00958-t001_Table 1Table 1Studies validating the clinical effect of tumor size and staging in TNM classification [18] for posterior uveal melanoma.Shields CL. et al., 2013 study (7731 patients) [15]Metastasis rate vs. Tumor size (T1–4)3 years:4% T17% T219% T332% T45 years:8% T114% T231% T351% T410 years:15% T125% T239% T363% T420 years:25% T140% T262% T369% T4Non-survival vs. Tumor size (T1–4)3 years:2% T14% T212% T318% T45 years:4% T18% T219% T330% T410 years:8% T113% T227% T343% T420 years:11% T124% T236% T351% T4Shields CL. et al., 2015 study (7731 patients) [21]Metastasis rate vs. TNM Staging3 years:2% S19.5% S225.6% S3100% S45 years:5.1% S116.7% S243.5% S3100% S410 years:11.9% S128.7% S261.1% S3100% S420 years:20.3% S143.8% S272.6% S3100% S4Non-survival vs. TNM Staging3 years:1.1% S15.3% S215.5% S3100% S45 years:2.8% S19.3% S227.2% S3100% S410 years:5.9% S115% S239% S3100% S420 years:8.3% S123.5% S253% S3100% S4Kujala E. et al., 2013 study (2152 tumors) [22]Survival vs. TNM Staging (S) At 5 years:96%Stage I89%StageIIA81% Stage IIB66% Stage IIIA45% Stage IIIB26% Stage IIICAt 10 years:88% Stage I80% Stage IIA67% Stage IIB45% Stage IIIA27% Stage IIIB10% Stage IIICAt 15 years:81% Stage I69% Stage IIA58% Stage IIB34% Stage IIIA18% Stage IIIB0% Stage IIICXu Y. et al., 2020 study (1142 patients) [25]Accumulative overall survival (OS) vs. TNM Staging88% for Stage I67.4% for Stage II46.3% for Stage III13.7% for Stage IVDisease-specific survival (DSS)Vs. TNM Staging93.6% for Stage I72.9% for Stage II49.5% for Stage III13.7% for Stage IVKapoor AG. et al., 2020 study on an Asian Indian patient cohort [26]Tumor size vs. 5-year distant metastases0% for T10% for T27% for T313% for T4 Tumor size vs. 5-year survival96% for T192% for T281% for T370% for T4The studies used either tumor (T) sizing or the staging (S) from the AJCC TNM classification [18]. The results support the importance of TNM classification for estimating metastatic risk and subsequent survival reduction.

Population-based cohort analyses conducted by Xu Y. et al. using the Surveillance, Epidemiology, and End Results (SEER) United States program reported a 5-year accumulative overall survival (OS) of 61.8% and a disease-specific survival (DSS) of 66.5% [25].

Kapoor AG. et al. studied the accuracy of the 8th edition of the AJCC Classification on Asian Indians and found later presentation with more T3 (39%) and T4 cases (26%) and a lower mean age of presentation [26]. Distant metastasis estimates at 5 years were 0%, 0%, 7%, and 13% for T categories T1–4, and the survival rates were 96%, 92%, 81%, and 70% [26]. These values were noted by the authors to be higher than those seen in Caucasian populations, as seen in studies such as Shields CL. et al. [15] and the AJCC Ophthalmic Oncology Task Force [17].

A study conducted by Khan S. et al. on iris melanoma estimated 5-year survival rates using the Kaplan–Meier method at 100% for T0 and T1a-c tumors, 90.4% for T2, 63.6% for T2a, and 50% for T3, T3a, and T4 [27] and an eight times higher non-survival rate from grade G2 to G3 tumors compared to GX and G1 [27] (Table 2).

Shields CL. et al. performed an analysis on 432 iris melanomas and estimated the survival rates [28] (Table 2). The authors reported a small cohort for T3 with no metastasis or no evaluable data [28]. Shield CL. et al. noted that the use of the 8th edition AJCC classification resulted in few T3 category cases, insufficient for effective statistical analysis and recommended grouping T3 patients into the T4 category due to extra-scleral extension being especially significant in terms of prognosis [28].

#### 3.1.4. Other Histopathological Findings with Prognostic Impact

Ki-67: Mlecnik B. et al. have reported prognostic value in evaluating the density and immune cell location within the tumor [29]. They reported their findings on using Ki-67 protein as a marker of proliferation and M30 as a maker of apoptosis and suggested that primary tumors could exhibit resistance to immune-mediated attacks [29]. In effect, high Ki-67 is a negative prognostic factor [30] since Ki-67 antibody positivity is lower in well-differentiated tumors and higher in less-differentiated, ectodermal stem-cell-like neoplastic uveal melanoma cells [30,31]. Together with other genetic criteria, higher Ki-67 positivity is associated with a tumor being classified as having a worse prognosis [31].

Lymphocytic infiltration in uveal melanoma is not as frequent as in cutaneous melanoma [30,32]; unlike other neoplasms (cutaneous melanoma, non-Hodgkin’s lymphoma, non-small-cell lung cancer, and breast cancer [32]), where the presence of inflammation could be associated with an enhanced host immune response and a likely better prognosis, in uveal melanoma, the inflammatory phenotype is associated with the loss of one chromosome 3 (monosomy 3) and, thus, with a negative prognosis [32]. This inflammation corresponds to tumors comprised of more epithelioid cells with higher mitotic activity, and is thus associated with greater chromosomal abnormality, such as in the case of monosomy 3 [32]. Broggi G. et al. reported a threshold of >100 lymphocytes per 20 40× high-power fields as being a negative prognostic factor [30].

Microvascular features: Foldberg R. et al. studied the microvascular features of uveal melanoma tumors and reported that the association between microvascular abnormalities, such as closed vascular loops with epithelioid cells, and high mitotic activity is detrimental to survival [33]. Foldberg R. expanded on the original research by using Factor VIII-related antigen (F8) to obtain a different histochemical image [34] compared to previous efforts using periodic acid-Schiff stain [33]. According to Foldberg R., F8 allowed for the separation of the influence of autofluorescence on the initially described vascular loops and network patterns [33]. Furthermore, F8 highlighted microvascularity in otherwise PAS “areas of silence” [34]. This study narrowed down five histopathological patterns with prognostic value: the presence of arcs, arcs with branches, loops, networks, and the absence of normal patterns [34]. These patterns can also intertwine, such as in the case of a network pattern, which also implies the presence of loops [34]. In total, 70% of the variation in the data was accounted for by the absence of normal circulation, the presence of arcs, and arcs with branches. Parallel or parallel crosslink vascularization was found in a larger study conducted by Foldberg R. et al. to be significant [33] and not significant in the follow-up Foldberg R. study [34]. Finally, the presence of a good prognosis with normal vascular pattern means a significantly lower microvessel count versus an unfavorable prognosis with a high microvessel count [34] and other vascular abnormalities.

### 3.2. Molecular Diagnosis

#### 3.2.1. Molecular Diagnosis Is the Next Steppingstone in Obtaining More Prognostic and Therapeutical Accuracy for Uveal Melanoma

Overall, the 7th and 8th editions of the AJCC Classification have been proven to accurately reflect metastatic disease and survival rates in large multi-center studies. However, as will be presented, genetic analysis of the tumor is also an integral and indivisible part of uveal melanoma (UM) since UM metastatic risk and prognosis are especially dependent on genetic profiles.

Fine-needle aspiration biopsy (FNAB) can be used to sample uveal melanoma for genetic deoxyribonucleic (DNA) or ribonucleic acid (RNA) profiling before commencing radiotherapy [2]. Other methods, such as transvitreal biopsy, can be employed for tumors located more posteriorly [35]. The histopathological analysis of enucleated specimens of larger tumors should always be accompanied by cytogenetic testing [36,37].

Genetic profiling can be employed for risk stratification. Table 3 chronologically summarizes pioneering research into the clinical impact of uveal melanoma cytogenetics by presenting several earlier studies evaluating chromosome 3 monosomy (M3), chromosome 3 disomy (D3), and chromosome 8q status conducted by Prescher G et al. [38,39], Onken MD. et al. [31], Damato B. et al. [36], Damato B. and Coupland SE. [37], Shields CL. et al. [39,40], and Ewens KG. et al. [41,42].

Table 3 also presents later studies from Shields CL. et al. [43,44] and Dogrusöz M. et al. [45] (Table 3), which consolidated the need for a personalized prognosis based on uveal melanoma cytogenetic data [44] and for integrating cytogenetics as an adjuvant to AJCC TNM classification [45].

The Cancer Genome Atlas (TCGA) classification, published in 2018 [5,46,47], uses chromosome 3 and 8 mutations to assess metastatic risk for all uveal melanoma types (choroidal, ciliary body, and iris) [5,46,47] by classifying them into four classes based on prognosis: Class A, B, C, and D [2,4,5,42] (Table 4).

#### 3.2.2. Genetic Profiles Are Intrinsically Linked to Uveal Melanoma Metastatic Patterns

Supporting the genetic role of metastatic patterns in uveal melanoma is Dogrusöz M. et al.’s previous work on TCGA classification [45]. The authors noted that in their study, the only patients with AJCC Stage I tumors that died were the ones with the most unfavorable genetic profile of combined monosomy-3 (M3) and 8q gain [45]. They suggested that this genetic makeup might be necessary for small tumors to metastasize [45]. Patients with AJCC Stage II and III tumors with only monosomy 3 or separately with only 8q gain did not succumb due to uveal melanoma metastases [45]. The authors also noted that a larger tumor size could have an additive effect on the accumulation of genetic pleomorphisms in the tumoral cell population; this effect was present in the monosomy-3 and 8q gain group, with an AJCC staging effect on non-survival being even more apparent [45]. Thus, genetic profiling and tumoral size, such as largest linear basal diameter of the tumor (LBD) (Corrêa ZM., Augsburger JJ.) [2,43] or the AJCC classification (Dogrusöz M. et al.) [2,41], can be independent prognostic factors [2,45,48], and the combined AJCC + chromosome data could be especially valuable for metastatic prognosis and overall survival rates [2,45].

The TCGA findings concerning chromosome 3 and 8 mutations, such as disomy 3, monosomy 3, disomy 8q, and 8q gain, were successfully validated in a large 10-year study reported by Vichitvejpaisal P. et al. at the Wills Eye Hospital in Philadelphia on samples obtained via FNAB [5], with 5-year cumulative rates for distant metastasis reported as being 4% for class A, 20% for class B, 33% for class C, and 63% for class D [5] (Table 4). TCGA classification was found to be a better predictor of metastatic risk and prognosis by Shields CL. et al. [2] and Mazloumi M. et al. in a comparative study [4]. Robertson AG. et al. [46] showed that poor-prognosis monosomy-3 uveal melanoma is associated with a distinct global methylation pattern [46] with BAP1 aberrancy that may result in a metastasis-prone deoxyribonucleic acid (DNA) methylation state [46]. Table 5 summarizes prognosis and metastatic risk for each class of the TCGA classification.

Mazloumi M. et al. compared TCGA to the established AJCC clinical-based model and reported the superior predication of metastasis risk and time to metastasize [4]. Interestingly, other variables, such as tumor thickness, largest basal diameter, and ciliary body involvement, showed value in the prediction of distant metastasis [4]; however, they were inferior predictors to the genetic classes of TCGA [4]. Furthermore, in their study, Mazloumi M. et al. did not find a statistical correlation for extraocular extension as a predictor of metastasis [4]. This would indicate that genetics play a strong role in the metastatic patterns in uveal melanoma.

Gill VT. et al.’s 2023 study proposed the incorporation of an improved classification system based on patient age, gender, AJCC T-categories, and TCGA groups A to D [49]. This study also highlighted the statistical significance of ciliary body involvement (CBI) in prognosis. Concerning therapeutic implications, Gill VT. et al.’s study reported the absence of metastasis for the first favorable prognostic class, while Group 4 patients had a very high metastatic risk, and this may warrant inclusion in randomized trials with adjuvant treatment [49].

#### 3.2.3. Genetic Drivers for Oncogenesis, Micrometastasis, and Metastasis


*Identifying Genetic Oncogenesis*


Onken MD. et al.’s genetic analysis of uveal melanoma filtered 3075 genes in order to discriminate tumoral classes [31]. The authors identified 62 genes of interest. Class 1 was considered the de facto favorable prognosis class with differentiated cells. The genetic analysis was focused on delineating genes responsible for class 2, which are less-differentiated tumors with higher metastasis probability and overall unfavorable prognosis. Class 2 genes included significant gene clusters on chromosome 3 (*p* = 0.002) that were down-regulated and on chromosome 8q (*p* = 0.004) that were up-regulated [31]. Gain on chromosome 6p, which is considered favorable to prognosis, was observed only in class 1 tumors [31]. Overall, the more common cytogenetic changes detected in uveal melanoma were loss of DNA on chromosome 3, gain on 6p, loss on 6q, and gain on 8q (3,16,17) [31], with loss of 6q and gain of 8q being the most prevalent [31]. Among the genes incriminated in class 2 tumors (worse prognosis class), the authors found that the top 26 of these genes’ functions were cell communication (13 genes), development (11 genes), cell growth (seven genes), cell motility (four genes), and cell death (three genes). Of the development genes, most have been implicated in neural crest development (which gives rise to melanocytes) such as Receptor Tyrosine Kinase (KIT proto-oncogene), Erb-B2 Receptor Tyrosine Kinase 3 (ERBB3), Endothelin Receptor Type B (EDNRB), Secreted Phosphoprotein 1 (SPP1), Frizzled Class Receptor 6 (FZD6), Transcription factor AP-2 Alpha (TFAP2A), and Spinal cord-derived growth factor (SCDGF-B) [31]. Most of these genes also present associations with other neoplastic processes. Receptor Tyrosine Kinases (RTKs) are single-transmembrane receptors that participate in the development and progression of a variety of tumors, promoting malignant behavior [50]. Overexpression (such as for ERBB3) is a clinicopathological characteristic in cutaneous melanoma [50]. In uveal melanoma, endothelin receptor B expression correlates with monosomy 3, changes to chromosome 8q (at position 24.1) and the presence of epithelioid cells [51], which have all been identified as being detrimental prognostic factors for uveal melanoma. SPP1 is a many-cancer oncogene [52], including melanoma [52,53], potentially promoting proliferation, migration, and invasion and inhibiting apoptosis [52,53]. Bromodomain and extra-terminal domain (BET) inhibitors of SPP1 could be beneficial in melanoma therapy [54,55] by inhibiting cell proliferation, migration, and invasion in an SPP1-dependent manner [54]. Fodor K. et al. reported a correlation between FZD6 and Vascular Endothelial Growth Factor A (VEGFA) [56], thus linking cancer stem cells (CSCs) in uveal melanoma to the angiogenic process [56]. TFAP2A encodes Activating Protein 2 Alpha (AP-2α), which facilitates melanoma metastasis through the transcriptional activation of genes within the E2F pathway, which inhibits apoptosis [57,58]. SCDGF-B has angiogenic activity similar to the Platelet-derived growth factor (PDGF)/Vascular Endothelial growth family (VEGF) superfamily [59]. Overall, class 1 generally corresponded to lower-grade spindle melanomas, whereas class 2 corresponded to higher-grade uveal melanomas with more epithelioid cells [31]. The genetic classification corresponded poorly with melanoma cell type, as recorded on the pathology reports [31], corresponding well with the findings related to cytologic rank [31] and offered superior prognosis information concerning both.

Onken MD. et al. followed up on their previous study [31] by measuring Ki-67 protein immunohistochemical marker’s prevalence, and they found increased Ki-67 positivity in class 2 tumors and in tumors with chromosome 3 loss and increased aneuploidy. Mlecnik B. et al. suggested that Ki-67 protein is a marker of proliferation and that primary tumors could exhibit resistance to immune-mediated attacks [29]. Onken MD. et al. suggested that more rapidly proliferating tumors are more likely to acquire chromosomal mutations [60]. Ki-67-positive tumors were more likely to metastasize; however, the immunohistochemical marker by itself was a less significant prognostic factor than genetic profiling and classification [60].

Less frequent than chromosome 3 loss and 8q gain are mutations in chromosome 1 (1p) [61,62] and the loss of chromosome 8p [41]. Chromosome 1p and 8p loss poses an increased risk of metastasis [41]. Chromosome 8p loss (initially defined by Onken. et al. as deletion of 8p12-22 and DNA hypermethylation of the retained hemizygous 8p allele region [63]) in the presence of chromosome 3 loss is considered a more important prognostic indicator for metastasis than chromosome 8q gain [41,63]. Thus, class 2 tumors with the highest metastatic risk would be those with chromosome 3 loss, 1p loss, and 8p loss [41].


*Oncogenic Initiation—First Genetic Driver Event*


Initiating mutations in either Guanosine Nucleotide-binding Protein Q gene (GNAQ) [64,65,66,67] or its paralogue [65], Guanosine Nucleotide-binding Protein Alpha-11 Gene (GNA11), act as a paramount driver in the oncogenic process [64,65,66,67,68,69] and are present in 80–90% of uveal melanomas [64,65,66]. GNAQ is located on chromosome 9q21.2 and GNA11 on 19p13.3 [65], which are paralogous genes with a sequence homology of about 90% [65] and a seven-exon coding region [65]. GNAQ and GNA11 mutations are mutually exclusive [64,65,66]. GNAQ and GNA11 genes encode protein members of the q class of G-Protein Alpha subunits, which are implicated in the mediating signals of G-Protein-Coupled Receptors (GPCRs) [65] and downstream effector [65,70] growth factors [66]. Other oncogenic driver mutations are in the downstream effector of G-Protein Alpha signaling PLCB4 (Phospholipase C Beta 4) [71,72] and mutation in Cysteinyl Leukotriene Receptor 2 (CYSLTR2)-encoding p.Leu129Gln substitution [72]. GNAQ, GNA11, PLCB4, and CYSLTR2 mutations lead to the constitutive activation of GPCR signaling [65] and the downstream activation of the Mitogen-Activated Protein Kinase (MAPK) pathway [10,72] (such as the activation of Protein Kinase MEK and MEK-ERK1/2 pathway [73]) and PI3K/Akt/mTOR, YAP/TAZ, Wnt/β-catenin, Rac/Rho, Notch pathways [73,74,75,76,77].

The aforementioned GNAQ, GNA11, PLCB4, and CYSLTR2 mutations with subsequent G-Protein Alpha pathway activation have been summarized by Rodrigues M. et al. as the first oncogenic steppingstone event [78]. Russo et al. proposed that inactivating mutations of the cyclin-dependent kinase inhibitor 2A (CDKN2A) through promoter methylation or the loss of the 9p region could be involved in triggering the metastatic progression of uveal melanoma [64].


*Oncogenic and Metastatic Progression—Second Genetic Driver Event*


According to Rodrigues M. et al., the progression of uveal melanoma relies on a second genetic driver event [78], which could be either of the following:(1)Bi-allelic inactivation of BAP1 (about 60% of cases) on chromosome 3p21 [78];(2)Change-of-function heterozygous mutation in Splicing Factor 3b Subunit 1 (Spliceosome Factor SF3B1) (about 25% of cases) [78];(3)Heterozygous mutation in Eukaryotic Translation Initiation Factor 1A X-linked (EIF1AX) [76] (about 15–17% of cases [78,79]).

Loss of chromosome 3 and mutations of the Ubiquitin Carboxyl terminal Hydrolase gene (BAP1) are strictly related to the progression of uveal melanoma [64]. BAP1 functions as a tumor suppressor [64] and is associated with poor prognostic factors (class 2 genetic profile) and higher metastatic risk [79]. Splicing Factor 3b Subunit 1 (Spliceosome Factor SF3B1) mutations are encountered in 25% of uveal melanomas and present intermediate metastatic risk [80], with a third of the patients developing metastatic disease at 5 years from the time of diagnosis [80]. Drabarek W. et al. validated that the low expression of Abhydrolase Domain-Containing 6 Acylglycerol lipase gene (ABHD6) could be used as a biomarker to select high-risk SF3B1 uveal melanomas [80]. EIF1AX mutations were associated with the class 1 gene expression profile (GEP) and the absence of ciliary body involvement [79]. Mutations in SF3B1 and EIF1AX were also almost mutually exclusive [79].

Compared to cutaneous melanoma, uveal melanoma does not present with B-Raf Protein gene (BRAF) mutation [59,74] of the MAPK/ERK signaling pathway [81].

#### 3.2.4. Metastatic Behavior and Exhibition of Metastatic Pleomorphism to the Original Primary Tumor


*Metastasis is not usually synchronous to the primary tumor*


The metastases of uveal melanoma are the most common cause of death in uveal melanoma patients [1]. However, only 1–3% of patients have evidenced clinical synchronous metastases at the time of diagnosis [1,23,82,83,84]. More than half (62%) of metastases are diagnosed within the first five years after treating the primary tumor [1,85], and the rest become clinically detectable even later [85]. Kujala E. et al.’s 25-year long-term analysis found that out of 289 treated patients, 145 (61%) died in the following 25 years due to uveal melanoma metastasis [85]. Considering only patients that died of uveal melanoma, an incredible 90% of patients died at 15 years, 98% at 25 years, and 100% at 35 years [85], despite having had received treatment for the primary tumors at the start of the follow-up.


*The “Seed and Soil” [83] for metastasis in uveal melanoma is not anatomic but genetic [23]*


Except for the direct transscleral invasion of conjunctival lymphatics [1,86], uveal melanoma disseminates in a solely hematogenous manner [1], often within the asymptomatic phase. Uveal melanoma has easy access to invade blood vessels (no basal membrane to breach); thus, circulating tumor cells can be found in uveal melanoma patients [1]. However, certain peculiarities are noteworthy. Firstly, certain genetic driver activations are required for metastatic activity [64,65,66,67,68,69,70,71,72,73,74,75,76,77,78,79,80], as exemplified by Dogrusöz M. et al.’s observation that only the patients with AJCC Stage I tumors that died were the ones with the most unfavorable genetic profile of combined monosomy-3 (M3) and 8q gain [45] and the crucial prognostic differences of the TCGA [5] and Onken MD. et al.’s class 1 and 2 classifications [31]. Secondly, despite anatomic access to closer sites, such as the lung, uveal melanoma overwhelmingly clinically disseminates to the liver first [1]. Thus, uveal melanoma adheres to the “seed and soil” theory historically postulated by Paget S. [1,87], in that certain organs are more suitable for specific cancers. However, this seed and soil for uveal melanoma is not anatomic; it is genetic. Primary tumor uveal melanoma cells express CXC Motif Chemokine Receptor 4 (CXCR4) and C-C Chemokine Receptor 7 (CCR7) [88,89]. The work of Li H. et al. found that CXCR4 and CCR7, which are both G-Protein-Coupled chemokine receptors, provided directional migration of uveal melanoma cells toward liver tissues [90]. CXCR4 is expressed in uveal melanoma cells, and it is known that ligand CXCL12 is expressed in the liver. CXCR4–CXCL12 interaction stimulates tumor cell migration and invasiveness and promotes the survival of CXCR4+ cells [90]. The CXCR4–CXCL12 connection provides an attractive explanation for the selective metastasis of uveal melanoma to the liver [88]. CXCR4 could provide early migration of uveal melanoma cells toward the liver [89]. Chemokine Receptor 7 (CCR7) has immunological functions in migrating naïve lymphocytes and mature dendritic cells to secondary lymphoid organs [88] and toward chemokines CCL19 and CCL21 [88]. CCL19 and CCL21 are ligands to CCR7 [88]. CCR7 expression in the primary uveal melanoma tumor was found by Van den Bosch T. et al. to strongly associate with poor survival [88]. Potentially, other liver metastasis interaction for uveal melanoma could be achieved through the c-MET Tyrosine kinase receptor [90] (MET Proto-Oncogene) [91] and Insulin-Like Growth Factor 1 Receptor (IGF-1R) [91]. c-MET expression is significantly higher in metastatic uveal melanoma tumors [92].


*Uveal melanoma exhibits metastatic pleomorphism; the metastasis can be cytogenetically different from the original primary tumor*


Primary tumor uveal melanoma cells express CXC Motif Chemokine Receptor 4 (CXCR4) and C-C Chemokine Receptor 7 (CCR7) [88,89]. According to Li H. et al.’s study, the chemokine receptor composition in uveal melanoma liver metastases suffers down-regulation after the metastasis of both CXCR4 and CCR7 [90]. The authors suggested that the liver microenvironment and soluble factors produced by hepatocytes led to a reduction in CXCR4 and CCR7 [90].

DNA methylation can be used to trace the tissue of origin of various tumors [6,93,94] such as central nervous system neoplasms [88] and melanoma [89]. Jurmeister P. et al. reported a uniquely characteristic methylation profile for uveal melanoma with distinct epigenetic signatures [6,95]. This methylation profile allows for the differentiation of uveal melanomas from melanomas of other primary sites [95]. Interestingly, Smit KN. et al. reported that the methylation profiles of BAP1 and SF3B1 mutations of uveal melanoma metastases differ from the original primary tumor [96]. The authors reported a plethora of epigenetic modifications in metastatic uveal melanoma and its metastases [96].

## 4. Discussion

Our understanding of uveal melanoma diagnosis and prognosis has steadily improved from histopathological to clinical and now to genetic. The early histopathological classification of Callender GR. [11] defined cell types. The Callender GR. and Armed Forces Institute of Pathology (AFIP) classifications allowed for the identification of the survival impact of epithelioid cell type [11] and of the link between increasing nuclear pleomorphism and decreased survival time [14]. The American Joint Committee on Cancer (AJCC) classification, which uses the universal tumor (T), node (N) and metastasis (M) (TNM) staging system for iris melanoma and posterior uveal melanoma involving the ciliary body and choroid [15,16,17] was a breakthrough in formulating prognosis in terms of uveal melanoma cases. Numerous studies have validated the effectiveness of the AJCC classification, such as Shields CL. et al.’s analysis of 7731 patients [15,21], Kujala E. et al.’s data for 2152 tumors [22], Smidt-Nielsen I. et al.’s retrospective study on 3344 Danish patients [23], Xu Y. et al.’s population-based cohort analyses using the Surveillance, Epidemiology, and End Results (SEER) United States program [25], Kapoor AG. et al.’s study on Asian Indians [26], Khan S. et al.’s study of iris melanoma [27], and Shields CL. et al.’s analysis of 432 iris melanoma patients [28]. Khan S. et al.’s study also looked at the impact of tumor grades and found an eight times higher non-survival rate from grade G2 and G3 tumors compared to GX and G1 [27]. The literature work converged toward cell class and genetic studies. The works of Prescher G et al. [38,39], Damato B. et al. [36], Damato B. and Coupland SE. [37], Shields CL. Et al. [40,41,42,43,44] and Dogrusöz M. et al. [45], Robertson AG et al. [46], Jager MJ. Et al. [47], and others were simplified and reunited under Cancer Genome Atlas (TCGA) classification [2,5,46,47]. Compared to the proven AJCC classification [15,16,17], the TCGA classification was found by Mazloumi M. et al. to be a superior predictor of metastasis risk and time to metastasis [4]. It became apparent that the genetic analysis of uveal melanoma was an important puzzle piece in predicting metastatic risk and survival.

Initiating mutations in GNAQ, GNA11, PLCB4, and CYSLTR2 mutations lead to the constitutive activation of GPCR signaling [65] and the downstream activation of the Mitogen-Activated Protein Kinase (MAPK) pathway [10,72] (such as the activation of Protein Kinase MEK; MEK-ERK1/2 pathway [73]) and PI3K/Akt/mTOR, YAP/TAZ, Wnt/β-catenin, Rac/Rho, Notch pathways [73,74,75,76,77]. This is a steppingstone oncogenic event for uveal melanoma. The progression of uveal melanoma relies on a second genetic driver event [78], such as the inactivation of BAP1 (about 60% of cases) on chromosome 3p21 [78], mutation in the Splicing Factor 3b Subunit 1 (Spliceosome Factor SF3B1) (about 25% of cases) [78], or EIF1AX mutation [78] (about 15–17% of cases [78,79]). Some of these mutations have a clinical impact in terms of genetic tumor classification (class 2 loss of DNA on chromosome 3, gain on 6p, loss on 6q and gain on 8q (3,16,17) [31]). The loss of chromosome 3 and mutations of the Ubiquitin Carboxyl terminal Hydrolase gene (BAP1) are strictly related to the progression of uveal melanoma [64].

Most uveal melanoma research focuses on primary UM despite the fact that metastases cause death in UM patients and not the primary tumor [97]. The primary tumor can be treated successfully using several options, such as enucleation [98], stereotactic radiotherapy [99], brachytherapy [100,101], and proton therapy [102]. Metastatic disease instead presents few realistic therapeutic options for improving survival. Furthermore, DNA methylation profiles of metastatic disease can differ from the original tumor [96], signaling genetic difference and pleomorphism. Smit KN. et al. argued that there has been a tendency to transpose treatments shown to be effective in CM to UM, such as immunotherapy and MEK inhibitors [97]. However, as described in multiple studies, the biological behavior of these two malignancies is completely different and, therefore, they require different approaches [97]. Outstandingly, uveal melanoma’s metastatic behavior is in complete opposition to cutaneous melanoma [90], with a strong tendency to metastasize toward the liver, thus reinforcing the findings of genetic differences. Rossi E. et al. highlighted that uveal melanoma shows a very limited number of molecular lesions that drive progression to metastasis [91], and thus, their analysis allows for precise prognostication [91]; however, the development of UM therapy has not kept to the path of innovation [91]. Uveal melanoma can evade immune surveillance via multiple mechanisms, such as inhibitory checkpoint programmed cell death ligand 1 (PD-L1), cluster of differentiation 47 (CD47), and cluster of differentiation 200 (CD200) [103]. Rossi E et al. summarized that the activation of two different oncogenic pathways, MAP-kinase and YAP/TAZ [91,104,105], a low mutational burden [9,91,106], a subsequently low number of neo-antigens that could be recognized by the immune system [106], and a pro-tumoral infiltrate [91,107] make metastatic UM difficult-to-treat neoplasia [91,108]. We theorize that advancing our understanding of the particular genetics of uveal melanoma is key to obtaining novel information. For example, while no systemic therapy received regulatory approval for patients with metastatic uveal melanoma until January 2022 [108,109], when Tebentafusp was approved by the United States Food and Drug Administration [109], this is merely similar to the beginning of genetic understanding; given a multidisciplinary approach and even more genetic research into uveal melanoma, progress could be made.

## 5. Conclusions

Our understanding of uveal melanoma has improved step-by-step from histopathology to clinical classification to the more recent genetic understanding of oncogenic initiation and progression. This review summarized knowledge on pathological diagnosis and molecular diagnosis; however, the practicality is in the fact that these intertwine to create a more complete clinical and prognostic picture of uveal melanoma. The current clinical challenge is represented by prognosing and treating metastatic disease since few metastases are synchronous with the primary tumor and most develop over time after primary treatment is completed. For these purposes, future research into the genetics of uveal melanoma could provide valuable information.

## Figures and Tables

**Table 2 diagnostics-14-00958-t002:** Comparison between the studies of Khan S. [28] and Shields CL. [28] on iris melanoma.

Khan S. et al.’s 2011 study on 131 patients with iris melanoma [27]			
Metastasis rate vs. Tumor size (T1–4)	100% for T0 100% for T1 a–c 90.4% for T2 63.6% for T2a 50% for T3 and T3a 50% for T4		
Survival vs. Histological grade	Baseline = Grade 1 and unknown grade (GX) Grades 2 and 3 incur 8 times less survival probability versus baseline		
Shields CL. et al.’s 2018 study on iris melanoma (432 patients) [28]			
Metastasis rate vs. Tumor size (T1–4)	3 years: 0.5% T1 1.8% T2	5 years: 2.3% T1 9% T2	10 years: 4.9% T1 14% T2
	T3 not evaluable (insufficient cases)		
	12.2% T4	33.1% T4	Not evaluable for T4

Please note that Shields CL.’s 2018 study was performed using the 8th edition AJCC classification [28] while Khan. S.’s study was performed using an earlier AJCC classification [27].

**Table 3 diagnostics-14-00958-t003:** Chronological synopsis of several studies concerning the clinical and prognostic impact of the cytogenetic differences in uveal melanoma. Authors, date, and study findings are noted for each study.

Prescher G. et al.’s 1992 Study on 34 tumors [38]
-Reported the existence of two poor prognostic mutations in uveal melanoma: monosomy of chromosome 3 and multiplication (gain) of chromosome 8q material [35]. -Identified combined 8q gain and monosomy 3 as a poor prognosis subgroup.
Prescher G. et al.’s 1996 Study on 54 excised tumors [39]
-Monosomy-3 was strongly predictive of poor prognosis (*p* < 0.0001). -Only patients with monosomy-3 tumors developed metastases with a 57% metastatic rate and only a 50% relapse-free survival rate.
Onken MD. et al.’s 2004 genetic study on 3075 uveal melanoma genes [31]
-Landmark study identifying 62 genes of interest for uveal melanoma. -Stratifies uveal melanoma into two genetic classes with prognostic implications: -Class I favorable prognostic: no mutation; or only chromosome 6 gain; -Class II unfavorable prognostic: down-regulation of genes on chromosome 3 and up-regulation of chromosome 8 genes.
Damato B. et al.’s 2007 retrospective study on 356 patients [36]
-Correlated genetic abnormality with histological features: -Monosomy-3 (M3) associated with epithelioid cells, closed vascular loops, high mitotic rate; -8q gain associated with basal tumor diameter, closed vascular loops and high mitotic rate; -Found that M3 associated with 8q gain; -Found that M3 associated with high metastatic risk and death; -Recommended routine cytogenetic analysis for uveal melanoma patients.
Damato B., Coupland SE.’s 2009 publication [37]
-Supplements the original 2007 study with improved genetic testing methods by using multiplex ligation-dependent probe amplification (MLPA) for more granular results vs. earlier Fluorescence in Situ Hybridization (FISH). -Reports that partial deletions in chromosome 3 also carry prognostic risk. -Early use of neural networks to individualize metastatic risk analysis based on (1) genetic tumor typing; (2) clinical staging; (3) histological grading.
Shields CL. Et al.’s 2011 study on 500 patients [40]
-Complete Monosomy-3 was the most negative prognostic factor: -Complete M3 metastatic probability at 3 years was 0% for small tumors, 24.4% for medium-sized and 57.5% for larger melanomas; -Incidence of complete M3 increased with tumor size: 17% of small tumors, 27% of medium tumors and 41% of larger tumors.
Ewens KG. et al.’s 2013 study comparing the genetic profiling of 320 uveal melanomas [41]
-Analyzed impact on prognosis of chromosome 1, 3, 6 and 8 mutations. -Chromosome 3 loss (monosomy-3) most significantly associated with poor prognosis. -Chromosome 8: Initially 8q gain was 2nd associated with poor prognosis, however after adjusting for other variables (patient gender, tissue source, tumor basal diameter) 8q gain lost statistical significance; chromosome 8p loss remained statistically significant for poor prognosis, 2nd after chromosome 3 loss. -Thus, the study signals the negative effect of chromosome 8p loss on prognosis. -Chromosome 1p-loss not statistically significant after adjusting for other values. -No evidence of protective effect of chromosome 6p gain.
Ewens KG. et al.’s 2014 study comparing the genetic profiling of 63 metastatic uveal melanomas and 53 metastasis-free control cases [42]
Abbreviations used: Chromosome 3p-linked tumor suppressor protein—BAP1; Eukaryotic Translation Initiation Factor—EIF1AX. - 0× Lowest risk tumors (reference): disomy-3/BAP1-WT/EIF1AX mutations; - 10× risk: Disomy-3 mutation; - 13× risk: Monosomy-3 + EIF1AX-WT alleles. EIF1AX mutations decrease metastatic risk by 8× independent of presence of BAP-1 mutations or no BAP-1 mutations independent; - 40× risk (highest risk): BAP-1 mutations. 77% of tumors carrying BAP1 mutations metastasized. BAP 1 mutation was almost universally associated with monosomy-3: all BAP1-mutant tumors except one had monosomy-3. -Considering that disomy-3 is a protective mutation, the loss of one copy of chromosome 3 uncovers recessive BAP1 mutations, which are highly metastatic.
Shields CL. et al.’s 2017 cytogenetic study on 1059 patients [43,44]
-Increased tumor size leads to greater genetic pleomorphism [43]. -Individualized risk analysis based on 52 cytogenetic signatures [44]. -Greatest negative prognostic with: 8p loss, 8q gain and complete monosomy-3 [44].
Dogrusöz et al.’s 2017 study on 522 patients [45]
-Patients with AJCC Stage I tumors only died if they had the most unfavorable genetic profile of combined monosomy-3 (M3) and 8q gain. The authors suggested that this genetic makeup might be necessary for small tumors to metastasize [45]. -Larger tumor size had an additive effect to the accumulation of genetic pleomorphism in the monosomy-3 and 8q gain group [45]. -The study concluded that combining AJCC TNM [18] staging with chromosome data for chromosome 3 and 8q status yielded additional prognostic information.
Robertson AG. et al.’s 2017 genetic analysis identifying four molecular and clinical subsets in uveal melanoma [46]
-Study linked with development of The Cancer Genome Atlas (TCGA) classification -Four molecular subsets identified by specific genetic signatures: -Disomy-3 samples separated by into good and intermediate prognosis groups by transcription profile analysis; -Monosomy-3 separated into two poor prognosis classes A and B by differences in transcription profiles leading to distinct pathway features of hypoxia, signaling and proliferation; -Monosomy-3 (poor prognosis) presented BAP1 alterations (85%); -Contrary to other cancers in uveal melanoma, the better prognostic cytogenetic types associated reduced immune-mediated inflammation (disomy-3 tumors did not present CD8 T cell immune response), while the poor-prognosis monosomy-3 variety exemplified highly increased local inflammation and high immune-response in 30% of cases with marked CD8 T cell infiltrate (CD8A expression).
Jager MJ. et al., The 2018 Cancer Genome Atlas (TCGA) classification [47]
-Classes tumor into four categories based upon chromosome 8 and 3 mutations.
Vichitvejpaisal. P. et al.’s 2019 TCGA Classification validation study on 642 patients [5]
-Genetic tumoral classes strongly correlated with 5-year metastatic risk from 4% for lowest Class A to 63% in Class D.
Shields CL. et al.’s 2019 Comparative analysis of TCGA and AJCC Classifications [2]
-TCGA classification accurately predicted uveal melanoma prognosis. -TCGA classification better predicted metastatic risk versus AJCC classification.
Mazloumi M. et al.’s 2020 Comparative TCGA-AJCC study on 642 patients [4]
-The genetic TCGA classification was a stronger predictor for metastasis than factors pertaining to tumoral size or extension such as largest tumor basal diameter or ciliary body involvement. -Genetic testing more accurately offers prognosis in uveal melanoma. -TCGA classification can identify high risk patients for adjuvant therapy.

Each study is presented chronologically, including the author, date, and a short-summary of the study findings.

**Table 4 diagnostics-14-00958-t004:** The Cancer Genome Atlas (TCGA) classification of uveal melanoma [5,46,47].

	The Cancer Genome Atlas Class			
Genetic Results	A	B	C	D
Chromosome 3 mutational profile	Disomy 3	Disomy 3	Monosomy 3	Monosomy 3
Chromosome 8 mutational profile	Normal 8q	8q Gain	8q Gain	Multiple 8q Gains

The Cancer Genome Atlas (TCGA) classification of uveal melanoma: This table is reprinted from *Ophthalmology*. 2019 Oct;126(10):1445–1453, Vichitvejpaisal P, Dalvin LA, Mazloumi M, Ewens KG, Ganguly A, Shields CL. Genetic Analysis of Uveal Melanoma in 658 Patients Using the Cancer Genome Atlas Classification of Uveal Melanoma as A, B, C, and D., page 2, copyright 2019, with permission from Elsevier [5].Essential references for the TCGA uveal melanoma classification are also Robertson AG et al. 2017 “Integrative Analysis Identifies Four Molecular and Clinical Subsets in Uveal Melanoma” Cancer Cell. 2017 Aug 14;32(2):204–220.e15 [46] and Jager MJ, Brouwer NJ, Esmaeli B. paper “The Cancer Genome Atlas Project: An Integrated Molecular View of Uveal Melanoma” *Ophthalmology*. 2018 Aug;125(8):1139–1142. [47]; we hereby acknowledge these essential references for TCGA classification [46,47].

**Table 5 diagnostics-14-00958-t005:** Prognosis and metastatic risk summary by TCGA class [5,46,47].

	Prognosis and Metastatic Risk by TCGA Class [5,46,47]			
Study	A	B	C	D
Robertson AG. 2017 study [46]	Good prognosis	Intermediate prognosis	Poor prognosis	Poor prognosis
Vichitvejpaisal P. et al., 2019 study [5]				
5-year distant metastasis risk	4% at 5 years	20% at 5 years	33% at 5 years	63% at 5 years
5-year liver metastasis risk	Not specified	5.9% at 5 years	18.4% at 5 years	42.9% at 5 years

This table summarizes prognosis from Robertson AG. et al.’s 2017 study [46] and metastatic risk from Vichitvejpaisal P. et al.’s 2019 study [5].

## Data Availability

For the purposes of this review paper, the presented data pertains to the original individual author’s papers, cited accordingly in the references section.

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
