# Peer review of "Pathological and Molecular Diagnosis of Uveal Melanoma"

_diagnostics, 2024, doi:10.3390/diagnostics14090958_

Round 1
Reviewer 1 Report
Comments and Suggestions for Authors
Dear authors,
I appreciate your work on the review article on uveal melanoma and it is evident that you made an effort to consolidate the results of numerous studies and support them with relevant references
Author Response
Thank you for the valuable review feedback. In accordance with other review feedback, we revised the manuscript including readability improvements by tabulating data in section 3.1.3. and including a chronological table with summaries of the studies in section 3.2. We thank you for your review.
Corresponding Author,
Paul Filip Curcă
Reviewer 2 Report
Comments and Suggestions for Authors
The role of a systematic review is to present and synthesise the recent evidence. The paper presents a comprehensive overview, but lacks “readability”.
The sheer mass of data becomes cumbersome to read and compare, resulting in fatigue, and the article will not have the impact it deserves. This is of particular concern in sect. 3.2
It would be very useful have many more tabulated data which compares findings from different groups.
Given that the studies encompass a large span of time, and given the advances in technology and treatment, some indication of the dates of the studies would be useful without having to check in the reference section. Admittedly, this is more cumbersome in a numbered referencing system.
Author Response
Thank you for the valuable review feedback,
We have revised the manuscript hoping to improve readability. We have reduced text volume in section 3.1.3 and tabulated statistical data for better readability. Section 3.2. has been reorganized. A chronological table has been added for studies in 3.1.2 with study summaries. Results for metastatic patterns have been moved to section 3.2.2 including a prognosis table; genetic analysis information has been moved to 3.2.3 which concerns genetic data. Thank you for your review
Corresponding Author,
Paul Filip Curcă